# Fermentation Strategies to Improve Soil Bio-Inoculant Production and Quality

**DOI:** 10.3390/microorganisms9061254

**Published:** 2021-06-09

**Authors:** Maria Vassileva, Eligio Malusà, Lidia Sas-Paszt, Pawel Trzcinski, Antonia Galvez, Elena Flor-Peregrin, Stefan Shilev, Loredana Canfora, Stefano Mocali, Nikolay Vassilev

**Affiliations:** 1Department of Chemical Engineering, University of Granada, C/Fuentenueva s/n, 18071 Granada, Spain; mvass82@yahoo.com (M.V.); agalvezp@ugr.es (A.G.); elenagr08@gmail.com (E.F.-P.); 2The National Institute of Horticultural Research, 96-100 Skierniewice, Poland; eligio.malusa@inhort.pl (E.M.); lidia.sas@inhort.pl (L.S.-P.); pawel.trzcinski@inhort.pl (P.T.); 3Department of Microbiology and Environmental Biotechnology, University of Agriculture-Plovdiv, 4000 Plovdiv, Bulgaria; stefan.shilev@au-plovdiv.bg; 4Research Centre for Agriculture and Environment, Council for Agricultural Research and Economics, 00184 Roma, Italy; loredana.canfora@crea.gov.it (L.C.); stefano.mocali@crea.gov.it (S.M.); 5Institute of Biotechnology, University of Granada, 18071 Granada, Spain

**Keywords:** biofertilizers, biocontrol, submerged and solid-state fermentations, immobilized cells, mycorrhiza, optimization strategies

## Abstract

The application of plant beneficial microorganisms has been widely accepted as an efficient alternative to chemical fertilizers and pesticides. Isolation and selection of efficient microorganisms, their characterization and testing in soil-plant systems are well studied. However, the production stage and formulation of the final products are not in the focus of the research, which affects the achievement of stable and consistent results in the field. Recent analysis of the field of plant beneficial microorganisms suggests a more integrated view on soil inoculants with a special emphasis on the inoculant production process, including fermentation, formulation, processes, and additives. This mini-review describes the different groups of fermentation processes and their characteristics, bearing in mind different factors, both nutritional and operational, which affect the biomass/spores yield and microbial metabolite activity. The characteristics of the final products of fermentation process optimization strategies determine further steps of development of the microbial inoculants. Submerged liquid and solid-state fermentation processes, fed-batch operations, immobilized cell systems, and production of arbuscular mycorrhiza are presented and their advantages and disadvantages are discussed. Recommendations for further development of the fermentation strategies for biofertilizer production are also considered.

## 1. Introduction

One of the most dynamic and expanding fields of research during the last years is the production, formulation and application of biofertilizers and biocontrol agents. Here, we will present examples of the effect of different types of fermentation processes on the behavior of mainly biofertilizers, but also on biocontrol agents. According to Malusa and Vassilev [1], a biofertilizer “could be defined as the formulated product containing one or more microorganisms that enhance the nutrient status (and the growth and yield) of the plants by either replacing soil nutrients and/or by making nutrients more available to plants and/or by increasing plant access to nutrients”, while biocontrol agents are microorganisms and microbial substances, which protect plants from different harmful pests.

Plant beneficial microorganisms appear as an important alternative to chemical fertilizers and as a part of the sustainable agriculture and scientific efforts to develop healthier soil and food [2,3]. Microorganisms with industrial/commercial interest, which demonstrate plant beneficial properties, are mainly bacteria (*Bacillus*, *Pseudomonas*, *Rhizobium*, and *Azotobacter*, among others) and fungi (*Aspergillus*, *Penicillium, Trichoderma*, *Beauveria*, *Metarhizium*, *Clonostachys*, and mycorrhizal fungi) [4,5,6]. Generally, these microorganisms are involved in the decomposition of the soils’ organic matter, nutrient cycling and management of soil minerals, solubilization of insoluble nutrients, such as phosphate- or potassium-bearing materials, releasing of plant stimulating metabolites, such as phyto-hormones, and the suppression of plant pathogens [7,8,9,10]. It should be noted that many of these manifest more than one of the above activities, as they possess multifunctional properties [11,12]. However, it is also important to mention that only 10% of the soil microorganisms inhabiting zones surrounding plant roots can be cultivated, while the rest are declared uncultivable in standard media [13]. Bearing in mind the key role of soil microorganisms in sustainable agriculture and the concerns regarding crop quality and human health, as well as the problems of a growing human population in combination with climate change, the biotechnological approach in production and formulation of plant beneficial microorganisms should be in the focus of the scientific community [14,15,16]. Biotechnology offers a wide array of techniques leading to bioformulates: starting from selection of promising microbial strains, characterizing their morphological, physiological and biochemical properties, testing their activity under fermentation and soil conditions and, finally, formulating them into commercial products [15,16,17,18]. Scientific literature is abundant with studies on bioinoculants mainly oriented towards isolation and screening of beneficial microorganisms, but studies on fermentation processes and formulation techniques are still lacking [10,19]. Results of a recent analysis in the field of plant beneficial microorganisms suggest a more integrated view of soil inoculants, with a special emphasis on the inoculant production process, including fermentation and formulation processes, and additives [19].

In this short review, we will present a comprehensive analysis of the microbial mass-production processes, and the strategies for their optimization, applied in the field of biofertilizers/biopesticides development, namely the submerged liquid, the immobilized-cell-based and solid-state fermentations, mentioning also the specific methods utilized for the production of mycorrhizal fungi.

## 2. Submerged Liquid Fermentation

### 2.1. Single Batch Operations

Submerged liquid and solid-state fermentation processes are the main biotechnological techniques to obtain microbial biomass or spores, which are further formulated into commercial products (Figure 1). In general, fermentation is the art of mass-cultivation of microorganisms, in the majority of cases using specific media and controlled process parameters, such as temperature, pH, aeration, and, if necessary, additional feeding [20].

In the submerged fermentation process, microbial cells are grown homogenously in a bioreactor with liquid medium under agitation/aeration, using the medium nutrient components and releasing specific metabolites. The overall biotechnological process depends on the type/form of the final formulated inoculant product. At the end of the fermentation process, microbial biomass and/or spores can further be used to formulate a solid commercial product based on solid carriers [10]. Alternatively, liquid formulations of plant beneficial microorganisms can be formed by the addition of substances directly to the fermentation cells-bearing broth to ensure long storage life of the commercial products maintaining high cell numbers/mL and metabolic activity enhancement [21]. Fermentation broth is also rich in specific metabolites, some of them with growth promoting or biocontrol activity, which can be separated from the biomass and purified for direct application in soil-plant systems. Therefore, submerged fermentation processes should be optimized towards high cell/spore density and/or high metabolic activity for producing specific metabolites with plant beneficial properties [22]. The choice of the fermentation model, optimization of the medium components, process parameters, as well as the design of the bioreactor functioning seem to be the most important points in developing upstream strategies for achieving a high biomass production and high metabolic activity of bioinoculants. It is widely accepted that 10 g/L is the maximum biomass production obtainable in a liquid batch fermentation process; however, following the above mentioned strategies in some studies, dry cell weight ranging from 20 g/L to 300 g/L are reported with enhanced metabolite production [23].

One of the first steps in achieving high biomass and metabolite production by a given plant beneficial microorganism is to optimize medium composition. When studying soil microorganisms, well-known synthetic media are normally used, but they do not always lead to optimal results. It was suggested that media used for selection of microorganisms with a desired profile should not be recommended for biomass accumulation and specific metabolite production in larger volumes [24]. Various studies have been reported for optimization of culture media for biomass and spore production. Each one of the medium components affects the growth, sporulation, and metabolite production activity. Carbon and nitrogen sources are very important, as well as their cost and availability [25] bearing always in mind the production volume and substrate price and availability (Table 1). Glucose, sucrose, lactose and other simple sugars are easily assimilated by microorganisms, but other economically attractive media based on agro-industrial wastes are successfully experimented for industrial inoculant production [26]. Cheese whey, malt sprouts, and molasses were included in media composition and showed high suitability in large-scale production conditions with additional cell-protective advantages demonstrated in the formulation stage [27,28]. Glycerol was also proved as a potential substrate because of its abundance as a side product from the biodiesel production and cell protecting properties [21].

An important parameter of developing efficient nutrient media is the relation between carbon and nitrogen. Using cotton-seed or soy flours to increase the nitrogen content resulted in higher biomass production of *Metarhizium brunneum* in a shorter time period, with a simultaneous higher spore production [30]. Development of short fermentation processes and inexpensive and available C and N sources will decrease the high cost of production, which is accepted as the main factor limiting the commercialization of plant beneficial microorganisms. Phosphate is another macro-component of the fermentation media affecting biomass and metabolite production. As in natural conditions, inorganic sources of P are the most important part of the fermentation media with a strong growth-limiting effect [42]. Alexieva and Micheva-Viteva [43] reported that, in *B. subtilis*, phosphate starvation has been shown to stimulate 10- to 30-fold extracellular enzyme production, particularly alkaline and acid phosphatases. Low phosphate concentration or controlled P supplementation to the medium particularly in the production stage is necessary for organic acid production and simultaneously for rock phosphate solubilization [24]. Similar controlled addition of low concentrations of phosphate is advantageous for enzyme production and indole-3-acetic acid of P-solubilizing fungi and bacteria [32,44].

Another attractive strategy is to use optimized media, not only to achieve high biomass growth, but also to enhance the production of bioactive plant beneficial compounds such as siderophores, indole-acetic acid, and anti-microbial compounds, including antibiotics. For example, by manipulating growth-limiting medium components, organic acids or antibiotic production can be easily managed [22,24,42]. This approach is very useful, particularly in biotechnological processes aimed at production of fermentation liquids containing additional, simultaneously produced plant beneficial components ranging from organic acids, fito-hormones and volatile and antifungal substances, among many others [39,45,46,47,48]. Such final fermentation products, which resulted after medium optimization and fermentation process parameters (pH, temperature, air supply and agitation) are rich in biomass and high spore numbers, but also in metabolites important for plant growth and health and can be applied directly with double effect on plants. In recent years, cell-free fermentation liquids with strong phyto-stimulating and biocontrol properties are attracting a growing interest because their action does not depend on the soil micro- and macro-biota and the typical stress factors affecting living-cell biofertilizers [10,47,48,49,50,51]. Optimization of the fermentation medium may include introduction of additional components in order to increase the system productivity or control metabolite activity. For example, in a series of experiments it was demonstrated that the addition of biochar as a medium component increased phosphate dissolution by plant growth promoting *Aspergillus niger* [31,41]. The mechanism of this effect includes partial removal of fluoride, which can be found in rock phosphate and in a higher production of organic acids.

Therefore, the properties of the plant beneficial microorganisms and the optimized mass production fermentation systems are important components in biotechnological production of biofertilizers and biocontrol agents. The latter should be with low industrial production costs allowing further formulation development and adequate field application. Other biotechnological tools, such as co-cultivation of fungal microorganisms with possible synergic interactions between strains, can facilitate formulations that can overcome environmental constrains, such as drought, salinity, high temperature, etc. in comparison to practices when a single strain is used [52,53]. Preliminary studies on strain compatibility including timing of spore inoculation or inclusion of fermentation broth of one microorganism into the medium composition of another microorganism are needed to build a co-cultivation strategy [54]. A very detailed study on co-cultivation of *Beauveria bassiana* and *B. brongniartii* suggested that the two fungi are different enough to avoid a real competition, but under specific stimuli they can manifest a higher virulence [53].

### 2.2. Fed-Batch Operations

Another biotechnological method of bioinocula production could be the traditional single liquid batch fermentation operation model. Particularly the fed-batch mode of fermentation has been successfully experimented in biofertilizer mass production, although this approach has been more frequently used in other biotechnological processes. The fed-batch fermentation operation involves an intermittent feeding of substrate or DO-based and pH-based feeding to ensure a determined rate of consumption when the substrate concentration decreases to a minimum. In the field of plant beneficial microbial production, fed-batch fermentation is an efficient tool for reaching a sufficiently high biomass or high concentration of phyto-stimulating metabolites [22]. Compared to a batch culture method, which is currently widely used by the industry, the developed fed-batch culture method provides approximately 20-fold higher viable cell counts of the microorganism or one fed-batch cycle is equal to 20 single batches [55]. By manipulating the operational parameters and applying a fed-batch strategy, an increase of cell concentration of *Azotobacter chroococcum* was observed from 1.54 CFU/mL to 4.21 CFU/mL with a simultaneous energy reduction [56]. In addition, in conditions of fed-batch fermentation, the increase of some process efficiency values, such as yield coefficient and productivity, were reported compared with the simple batch operation [57]. In comparison to the simple batch cycle and depending on the feeding strategy, different biomass production rates and specific metabolic (normally plant growth promoting compound or biocontrol agent) productivity could be achieved as documented by Sharma et al. [58] using *Pseudomonas fluorescens*. Fed-batch strategy can be applied to solubilize insoluble inorganic phosphate by *A. niger* [59]. The phosphate was added four times at predetermined periods during the time-course of a batch process and then the liquid fraction was used as a soluble phosphate source. Organic acid production by the fungus was optimized during the first 48 h, before the addition of the phosphate material, thus increasing the P-solubilization efficiency. Fed-batch principles of fermentation were also successfully applied in *Bacillus thuringiensis* cultivation based on synthetic media improving cell concentration at least 50% [60] or more (from 6 g/L to 50 g/L; [61]) or spore concentration up to 1.25 × 10^10^ spores/mL [62]. Similar process improvement was achieved when sewage sludge was used as a substrate [29]. The bacterial spore concentration increased from 5.62 × 10^8^ to 8.6 × 10^8^ CFU/mL with a simultaneous entomocidal activity enhancement from 13 × 10^9^ to 18 × 10^9^ spruce budworm potency units per litre.

## 3. Immobilized-Cell-Based Fermentation Processes

The immobilization of cells can be defined as “the physical confinement or localization of cells to a certain defined region of space with preservation of some desired activity” [63]. An immobilized cell system is composed by three components: the cells, the matrix (carriers), where cells are immobilized, and the solution that occupies the rest of the matrix and may contain additives. The methods of immobilization are different, but mainly based on adsorption on solid carriers and (macro/micro)-encapsulation in gels. Immobilized systems are widely used in various biotechnological processes, but are still limited for agricultural purposes. In particular, immobilized cell technologies are used to study the behavioral changes in cells of plant beneficial microorganisms as the immobilized state is the normal state of microorganisms in soil [64]. These methods are also involved in formulation of biofertilizers [15,65]. As stated by some earlier studies and further confirmed by many other works, the immobilization of cells offers a number of advantages, such as a greater number of cells in one volume unit, the possibility of easier continuous fermentation process, greater metabolic stability and activity, easier downstream operations [66,67,68,69] and, particularly in the field of formulation of biofertilizers, the slow release of cells in soil-plant systems and the possibility to combine different types of bioeffectors in one product [70,71,72]. In fermentation conditions, immobilized cells are applied in processes for solubilization of insoluble phosphates or in production of metabolites with plant growth promoting effect. In the first case, organic acid producing bacteria and filamentous fungi are first immobilized in optimized conditions and then used as P-solubilizers [73]. The optimization process is necessary to improve both the efficacy of the immobilization procedure and metabolite (organic acid) production. In similar experiments, the one-factor-at-a-time scheme can be used for statistical experimental design and empirical modelling [74]. This approach was applied in animal bone char solubilization by *Aspergillus terreus* producer of itaconic acid [75] and in the production of cell-free liquid biofertilizer by *Piriformospora indica* in repeated batch modes of fermentation [21]. We suggested a number of biotechnological tools aimed at increasing the efficiency of the fermentation-based solubilization of insoluble phosphates by immobilized microorganisms [76,77], but despite some recent studies [78] there is no significant progress in this field.

Another possibility of wider application of immobilized-cell technologies in the production of biostimulants is to produce plant hormones (auxins, cytokinins, ethylene, gibberellins, and abscisic acid) by selected microorganisms [79] for further application in soil-plant systems. In this approach, gibberellic acid, which promotes plant cell growth, is an excellent example as its production in repeated-batch and continuous fermentation process by immobilized *Gibberella fujikoroi* has been documented more than 30 years ago when sodium alginate was found to form stable and firm beads with 2 to 3-fold higher productivity of the fungal cells entrapped inside compared to their free form [80,81]. More recent studies confirm the already known advantages of immobilized-cell based gibberellic acid production but also introduce some improvements in the carrier and substrate materials [82] or fermentation process bioreactor and design [83,84,85]. Another plant hormone produced by immobilized microorganisms is indole-3-acetic acid (IAA). It is involved in many plant activities, such as cell enlargement, cell division and tissue differentiation and determines plant behavior in different ambient conditions (light, gravity, etc.) [86]. Cells of *Klebsiella oxytoca* were found to produce IAA after immobilization by adsorption on inorganic carriers using sol-gel methodology and storage during various periods of time [87]. While the IAA production by the free cells decreased sharply after 30 days of storage, the results showed high stability of the immobilized cells and slight reduction of IAA production after 90 days of storage. IAA production was registered in a repeated-batch fermentation of *B. thuringiensis* cells entrapped in k-carrageenan [32]. In addition, during the fermentation process, solubilization of insoluble phosphates was carried out. Therefore, IAA and soluble phosphate were presented in the fermentation liquid, which potentially could be used as a plant growth promoting material. The immobilized *B. thuringiensis* cells demonstrated high stability, metabolic activity, and catalytic longevity in comparison to the free cell forms during at least five batch cycles. Using the above approach, immobilized cell technology could be applied in the production of microbial metabolites with high potential, such as harpin proteins or lipochitooligosaccharides (also known as Nod factors,) which also have found large acceptance as commercial products applied to improve plant health and symbiotic relations with nitrogen-fixing rhizobium bacteria or mycorrhizal fungi [88,89]. Immobilized cell technology was recently applied in the repeated-batch production of mycelium and spore material of *Clonostachys roseum* and *Metarhizium brunneum* for direct application or introduced/formulated into gel carriers (Vassilev et al., unpublished results).

A possible immobilized-cell-based production of cell-free liquids containing plant-beneficial metabolites could reduce the production process costs and, on the other hand, the final products will be free of all disadvantages related to the microbial survival particularly in harsh soil conditions. Similar advantages can be considered in accumulation of spores and biomass for further formulation of commercial products.

## 4. Solid-State Fermentations

Solid-state fermentation (SSF) is a fermentation process based on substrates in solid forms and carried out in the absence of free water [90]. This mode of fermentation has attracted the attention of many scientists because it is in fact a natural process, with high economic potential, which can be easily performed in laboratory and industrial conditions to produce various microbial products, including biofertilizers, while recycling residual agro-industrial materials [91]. During the last 20 years, a wide number of studies appeared on the utilization of SSF in the production of plant beneficial microorganisms [92]. Although an economic comparison between submerged and SSF fermentation processes has not been analyzed and published, it seems that SSF is widely accepted as an advantageous fermentation tool in biofertilizer and biocontrol production. In particular, the inclusion of agro-industrial waste products as media constituents ensures microenvironment conditions similar to that of soils (soil organic matter). On the other hand, the same products offer the physical supports (or carriers) for plant beneficial microorganisms [93], which facilitate the formulation process of biofertilizer production [18]. SSF can be successfully used in biocontrol production with the typical example of *Trichoderma* spp. grown on grape marc and wine lees [33] although cereals, straw, and wheat bran are also frequently used as substrates [34]. Other biocontrol fungi such as *C. roseum*, *M. brunneum*, and *Beauveria bassiana* are normally grown on cereal grains, such as rice, barley, rye, wheat, sorghum, or corn without any additional medium components [35]. Similarly, P-solubilizing fungi were successfully experimented in conditions of SSF using various substrates such as sugar beet wastes and olive mill wastes [36,37,38]. Further direct applications of the resulting products resulted in high plant growth and nutrition, enhanced biochemical soil activity, and resistance in desertified and metal contaminated soils [94,95,96,97].

SSF offers a unique interaction between air, solids, and the microorganism, which facilitates microbial metabolite production and spore formation. However, to achieve sufficient microbial growth, spore production, and high metabolic activity, optimization of the process parameters should be carried out. The most important factors affecting the microbial behavior are the inoculum volume, moisture, temperature, pH, addition of nutrients, aeration, and solid substrate particle size (surface/volume ratio). Optimization of medium could also enhance the overall productivity [98] or determine the need of additional substances able to modify the metabolic activity towards the desired amount of the final product [41].

It is important to underline that there are two types of SSFs depending on the mode of application of the final products. In the first case, SSF entire products can be applied directly introducing organic matter degraded by the microorganisms and the metabolic products released during the process [38,99]. Alternatively, spores formed due to nutrient limitation can be separated from the fermentation broth and further formulated as powder or granules applying different types of carrier materials. In general, SSF offers higher spore numbers and longer spore viability and efficacy compared to the liquid submerged process [100,101,102,103,104]. These two different types of applications obviously offer different benefits to soil-plant systems. In any case, what is most important is the final economic result, bearing in mind the whole production chain, including all upstream and downstream operations, as well as the functional longevity after different periods of time [18]. As an advantage of the SSF, the possibility of using the solid substrate, as a support/carrier, should be mentioned [18,92,105] as to what facilitates the formulation process. If some additional substances are used in the upstream stage with beneficial formulation characteristics, the overall advantageous profile of the SSFs is obvious. For example, if the solid substrate-moistening agent is glycerol, its cell-viability preserving traits can additionally increase the maintenance period of the final products [39,40]. As in the case of submerged fermentation mode, SSF is a process, which allows co-cultivation [106], and therefore, biomass of two beneficial microorganisms could be produced and further formulated or applied directly in soil-plant systems. However, it is important to test compatibility between experimental microorganisms before the fermentation trials [107].

## 5. Key Fermentation Parameters

Independently of the mode of fermentation (submerged or solid-state) or cell state (free or immobilized), we can always improve the process productivity by optimizing the initial pH, agitation rate, aeration volume, initial inoculum concentration, and (in SSF) initial moisture and type of the moistening agent (Figure 2). The assessment of the effect of different fermentation parameters on the biomass accumulation and cell survival in the bioformulates during storage is essential to the development of stable commercial products. The optimization schemes and experiments are normally carried out in controlled conditions simultaneously or by analyzing the effect of one parameter (variable) at a time, independently of the mode of fermentation (solid, liquid, free/immobilized cells, etc.).

The initial medium pH is important for the adaptation of the inoculum, affecting microbial growth, and, later, its metabolic activity and production of enzymes, antibiotics, organic acids, phytohormones and biocontrol potential as well. Microbial growth is strongly dependent by the pH as it affects the surface of cells and increases nutrient absorption [108]. Most microorganisms are reported to prefer neutral and slightly acidic pH with optimal biomass production in the range of 4.5 and 6.5. A pH value below 4 or above 7.0 slows fungal growth. Some species were found to tolerate alkaline pH while others, mainly some organic acid producers, preferred an acid environment [109]. For example, the export of citrate by *A. niger* was higher at medium pH of 2.0, and only 25% of that value was registered at pH 7.0 [110]. Other acid producers, however, need higher pH and the presence of a neutralization agent (CaCO_3_ or NaOH, Ca(OH)_2_) to produce the acid [109]. Many filamentous fungi, including some *Trichoderma* spp., decrease the medium pH during their growth, but after assimilation of the available carbohydrate substrate the opposite pH tendency is observed which influenced the biomass growth and enzyme production [111]. These details are important when the aim of the fermentation is to solubilize insoluble inorganic phosphate to enrich the final liquid with plant available P, to produce enzymes (for example cellulase) in the fermentation broth or when a bio-formulate is prepared for a soil-plant system to enhance the plant available P or degrade cellulose-bearing wastes.

Temperature is another important fermentation process parameter. There are plant beneficial microorganisms sensitive (or not) to high or low temperature but, in general, high temperatures facilitate the biomass growth. It was suggested that as the soil microbial community composition is affected by the climate change and freeze-thaw cycles, the effect of the temperature should be studied more in detail [112]. Formulated inoculants can be produced by air-drying, desiccation, lyophilization, and spray drying, among others [72]. Lyophilization (freeze-drying) is a soft dehydration method, which preserves the cell viability of the bio-formulations [113]. It was considered that fermentation pH and temperature affect the cell stress resilience and efficacy of this method [114]. Optimization of the fermentation parameters can be carried out including in the optimization scheme components of the medium [41,115].

The majority of processes for biomass/spores production of plant beneficial microorganisms are based on aerobic fermentations. In case of submerged fermentations particularly important is the dissolved oxygen as it is the main parameter for successful microbial growth, which directly affects metabolic activity and final products type. This phenomenon is more pronounced, specifically in fed-batch operations when the biomass concentration is varying. At laboratory conditions, the level of dissolved oxygen can be manipulated by the shaker speed, while using fermenters the oxygen dissolved in the medium is controlled by manipulating the airflow or the stirrer speed [116]. Applying different configurations of flow impellers plays an additional role in the dissolved oxygen level and also affects cell morphology, mass transfer, medium viscosity and rheology, which improved the product formation [117]. In solid-state fermentation, the use of special cultivation bags ensures gas exchange or alternatively, special designed bioreactors and oxygen-enriched air could be supplied, thus providing sufficient oxygenation necessary for optimal biomass accumulation and sporulation on solid-substrate fermentations [118]. Initial moisture and water activity, the nature and size of the solid particles, inoculum volume, and additional nutrients also affect the efficacy of the solid-state fermentation [92,98]. All these factors must be optimized applying strategies, which include factorial design and response surface methodologies, artificial neural network and genetic algorithm [119].

In some cases, the optimization strategies for fermentation parameters are oriented to shape the cell morphology [120]. If the aim of the fermentation is to produce an efficient, multifunctional product with high level of both biomass and metabolites, it shall be considered that the metabolic productivity is particularly dependent on an optimal morphology (free cells of pellets): another important critical parameter specially for fungal or filamentous microorganisms [121].

## 6. Selection of the Fermentation Mode for Industrial Production

The aim of the cultivation of plant beneficial microorganisms is to further prepare formulated commercial products based on biomass, spores and eventually the fermentation liquid, which contains plant-stimulating or biocontrol metabolites [10]. The selection of the fermentation mode can be determined experimentally as microbial biomass and spores from plant beneficial microorganisms can be produced by both submerged and solid-state fermentations. The first process, either as a single batch or fed-batch, is the most effective and accepted method for the production of biomass, spores, and metabolites on an industrial scale. On the other hand, the second one is instead a relatively well-established fermentation technology for the production of various metabolic products and significant amount of biomass and spores [116]. In liquid submerged fermentation processes, agitation and oxygen transfer rate are accepted as significant factors that affect both metabolite production and microbial growth. Indeed, for a successful fermentation process in a submerged process bioreactor, agitation is essential to achieve efficient heat and mass transfer, medium component homogenization, and a dissolved oxygen level to avoid biosynthesis of undesirable metabolites [122].

Solid-state fermentation, performed on a solid substrate with a low moisture content, offers a number of advantages, such as the use of economically attractive substrates, lower water and energy consumption, but with complicated (in some cases) downstream processing [92]. It is important to know that based on the water activity requirements fungi and yeasts are usually accepted as the most suitable microorganisms for solid-state fermentation. Fungi and yeast have lower water activity (a_w_) requirements, typically around 0.5–0.6 a_w_, while bacteria have a higher requirement (0.8–0.9 a_w_). In any case, the choice of the mode of fermentation for a selected microorganism should be also based on the type of the substrate used and the final metabolite product that has to be produced [119].

Another important consideration is the overall microbial activity and biomass and spores production as a function of the fermentation process mode. It is well documented that metabolite production is higher and can be achieved in a shorter time in solid-state fermentation when compared to submerged fermentation using the same medium [121]. Fungi grown in conditions of submerged fermentation have shown a greater sensitivity to catabolite repression compared to solid-state fermentation [122]. Spores produced in solid-state fermentations are characterized by higher stability and germination rate after freezing compared to spores obtained by submerged fermentations and are more resistant to dehydration, which facilitates their formulation [123]. In general, airborne spores formed under natural conditions are more stress tolerant (including UV radiation resistance) that those grown in liquid media. Solid-state fermentation conditions are closest to the natural environment [92]. Similar observations are reported for immobilized cell systems [64]. A great advantage of solid-state fermentation is also the possibility to directly apply the final product without special formulation procedures [16], although spores of plant beneficial fungi are easily separated from the mycelium-substrate mass and further formulated as powders or granules using various materials as carriers [72]. Talc powder, polyvinylpyrrolidone, vegetable/mineral oils, peat, starch-mannitol, methylcellulose, xanthan gum, vermiculite-bentonite, press mud from sugar production, and natural gel-forming materials (alginate, carrageenan, chitosan, agar) are widely used in seed treatments or direct introduction into soil-plant systems [92].

Therefore, when selecting the fermentation mode of plant beneficial microorganisms’ production, all this information should be evaluated in order to produce the most economical, environmentally friendly, stable and viable, easy to maintain in different storage conditions, and easy to apply commercial product.

## 7. The special Case of Mycorrhiza

Arbuscular mycorrhizal fungi (AMF) are an important group of soil microorganisms, which form beneficial symbiotic associations with bidirectional flow of nutrients between roots of wide range of plants and the surrounding soil microenvironment [124]. Mycorrhizas are also involved in biological processes, which improve plant health, through increased protection against biotic and abiotic stresses [125], improve soil quality and structure, and enhance nutraceutical value of horticultural products [126]. The most important advantage of using AMF as biostimulants is related with their role in facilitating phosphate and micronutrients uptake by plants [127]. While almost all plant beneficial microorganisms including some ectomycorrhizal and mycorrhiza-like can be produced by cultivation in fermentations systems, AMF cannot be mass-produced without plants [128]. The requirements for development and selection of bio-techniques for mass production of mycorrhiza are similar to those of other microbial fertilizers. The final product should be free of pathogens, viable after periods of storage, with a high colonization power and easy to apply [129]. The main problem in the case of AMF is the lack of both free-of-host efficient production methods and formulations that cannot ensure high final product quality [128]. AMF are commonly produced in scaled-up pot plant-based culture using sterilized substrates. However, this method is not economically acceptable in industrial conditions. Soilless, hydroponic culture is another mode of AMF production as it provides high quality inoculum with higher spore number compared to the pot cultivation in greenhouse [130]. In both modes of production, inert materials such as bark, calcinated clay, expanded clay and perlite are used, alone or in combination, in the medium composition as they are with stable characteristics, facilitate aeration, and hold sufficient water for plant growth [129]. Other biotechnological approaches to improve the AMF mass production in conditions of soilless cultures are using nutrient film technique, aeroponics, and root organ culture [130,131,132,133]. It is interesting to mention the almost equal price per spores produced by these techniques with some advantage of root organ culture [134]. Using the split-plate technique of St-Arnaud et al. [135] it is possible to repeatedly remove and replace the media in the mycorrhizal side of the plate and thus continually harvest spores and hyphae from the system [45,136]. In 2000–2001 AMF were cultivated using alginate and k-carrageenan as media constituents (Vassilev et al. unpublished data). A similar approach could be used with other natural polysaccharides. However, this process is slow and time-consuming, the substrate is expensive, thus economically unsustainable, and seems unlikely to be adopted by the industry.

Like in many other biotechnologically based productions of agricultural, plant beneficial products, a strong cooperation between industrial microbiologists, plant physiologists, agronomists, and soil scientists are needed to develop an inexpensive, efficient, and clean technology for the AMF production. Further progress in this field could be expected in the formulation part of the mycorrhizal inoculum production, which could include mycorrhiza-helper bacteria and microorganisms with other functions, such as nitrogen-fixing, P-solubilizing, or biocontrol functions. Similar attempts have been published earlier with promising results [64].

## 8. Perspective Developments and Conclusions

Microbial fertilizer production comprises a series of operations including mass production of the target microorganism in fermentation systems (upstream stage), followed by product formulation. In some cases, the product is directly applied after the production stage. In any case, the choice of the fermentation mode is of great importance as it determines the quality and amount of the useful microbial mass and metabolites promoting plant growth and health. Biotechnology offers a number of well-developed fermentation schemes, which can fulfil the desired final results, depending also on the formulation schemes. Plant beneficial microorganisms should be easily and economically produced by the fermentation industry. The main point from an economical point of view is to ensure reasonably low-priced substrates, available locally and with stable seasonal characteristics. Despite the recent progress in the field of production and formulation of biofertilizers, additional studies are needed to further improve the upstream operational productivity using other fermentation strategies such as, a continuous mode of fermentation or different configurations of the bioreactor. Here, the use of immobilized cell systems should be more actively studied as an alternative technique. Optimization of methods should be widely applied to establish highly efficient upstream processes focusing on optimized media and parameters. Special attention should be paid to AMF production in fermentation systems, as they have a market potential ready to be exploited due to farmers’ acknowledgement of their role in soil-plant systems development.

New research for the optimization of both fermentation and formulation processes would be necessary in the future for biofertilizers based on other kinds of microorganisms such as yeasts, since isolates from genera such as Williopsis, Saccharomyces, Candida, Meyerozyma and Pichia have been shown to promote plant growth and nutrition with different crops [137,138]. Indeed, even though yeast production is a well-developed industrial sector for the wine and bread industry, the application in field crops poses some challenges that have to be addressed. The exploitation of processes fostering the production of metabolites could also be useful to induce expression of specific metabolites necessary for the colonization of roots or enhancing the effect of the microorganisms (e.g., lipochitooligosaccharides) in enabling endosymbiosis [139]. However, the production of biofertilizers based on non-obligate endosymbiont mycorrhizal fungi of the order Sebacinales as well as of other endophytic microorganisms would also require a specific development of fermentation and, most of all, formulation processes aiming at improving the colonization of plant tissues.

Optimizing the fermentation process in view of reassembling strains with differing modes of action into small communities, thereby providing more consistent protection or growth promotion than with the application of single strains, is a challenging goal worthy to venture as such strategy could widen the application of bioproducts [140,141]. Further development of the fermentation strategies for biofertilizer production should also be focused on products able to manipulate and control phyto-microbiome structure [142], which will be the next biotechnological approach in the Sustainable Agriculture.

However, the research aiming at optimizing the strategies for fermentation and formulation of bioinocula must take into consideration the economic, environmental and societal features in order to comply with sustainability requirements. In this respect, since the production cost is among the constraints expressed by farmers for the usage of bioinocula products, possible ways to reduce it could be searched by testing new materials for the fermentation process. For example, the rapid development of the biogas industry, which produces tons of biodigestates difficult or costly to deploy due to environmental constraints, could also provide ingredients for cheap fermentation substrates [143]. SSF could also exploit the use in the fermentation medium of carbon-based materials which could substitute to a large extent other inputs, useful to produce multifunctional inocula [144]. Such approaches would effectively comply with sustainability conditions.

Even though, the market potential of microbial-based products was confirmed by recent analyses, which valued at about 10.2 billion USD by 2025 the global biopesticide market, with an annual growth rate of approximately 15% [145] and projected 3.15 billion USD by the end of 2026 for the biofertilizers market, at an annual growth rate of about 11% [146], the regulatory framework could pose some bottlenecks for the sector development, including the fermentation and formulation phases. Indeed, as an example, in case of biofertilizers, the new rules enacted in the European Union in 2019 [147] are currently limiting the marketing to just four kinds of biofertilizers, for nitrogen supply (symbiotic Rhizobium spp. and free-living Azotobacter spp. and Azospirillum spp.) and for phosphorous nutrition (mycorrhizal fungi). Furthermore, only the drying or freeze-drying processes in the formulation of the product are allowed, which is also restrictive considering the technologies available. Nevertheless, the improvement of fermentation strategies would be useful to comply with quality and safety requirements of bioinocula, which are the basis to increase their use in the field and thus support the transition toward a new agricultural “bio-green revolution”.

## Figures and Tables

**Figure 1 microorganisms-09-01254-f001:**
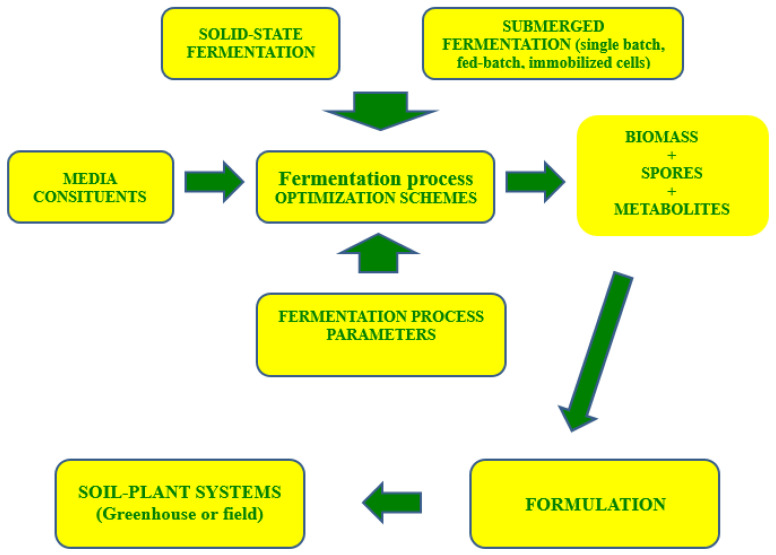
General scheme of submerged and solid-state-fermentation-based production of bioformulates.

**Figure 2 microorganisms-09-01254-f002:**
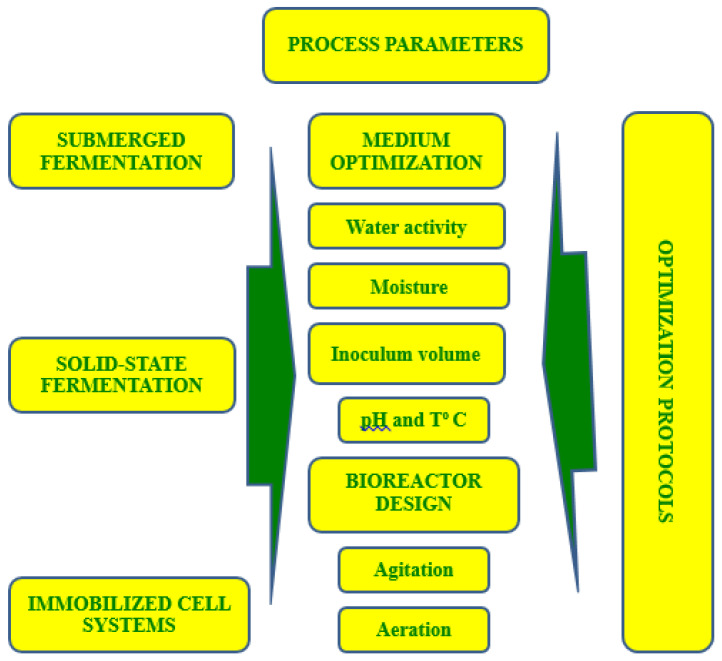
Optimization options for different fermentation profiles.

**Table 1 microorganisms-09-01254-t001:** Overview of materials used as substrate sources or additives for the development of strategies with different fermentation technologies.

Fermentation Mode	Utilization Mode	Materials Used	References
Liquid submerged	Substrate source	Glycerol	[21]
		Agro-industrial wastes	[26]
		Cheese whey	[27]
		Malt sprouts	[28]
		Beet and cane molasses	[24] and ref. therein
		Sewage sludge	[29]
	Substrate additive	Cotton-seed	[30]
		Soy flours	[30]
		Biochar	[31]
Solid State	Substrate source	Corn steep liquor	[32]
		Grape marc and wine lees	[33]
		Straw	[34]
		Wheat bran	[34]
		Cereal grains (rice, barley, rye, wheat, sorghum, corn)	[35]
		Olive mill solid wastes	
		Sugar cane bagasse	[36,37,38]
		Sugar beet wastes	[36]
		Vinasse	[31]
			[24] and ref. therein
	Substrate additive	Glycerol	[39,40]
		Biochar	[41]

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
