# Peer review of "Fermentation Strategies to Improve Soil Bio-Inoculant Production and Quality"

_microorganisms, 2021, doi:10.3390/microorganisms9061254_

Round 1

Reviewer 1 Report

The submitted manuscript represents information about Fermentation strategies to improve soil bio-inoculant production and quality. The topic is very interesting. The introduction section is well written and all aspects regarding the main topic are described. The authors have made a very interesting statement information. However, the review work requires not only an interesting text, but also must have various graphic or tabular forms of data presentation, which increases the interest of readers and allows easier understanding of the conveyed content.

Author Response

Thank you for your suggestions and critical remarks. In additions to the Figures, a Table was included highlighting  some important characteristics of the fermentation processes.

Reviewer 2 Report

I reviewed the article entitled "Fermentation strategies to improve soil bio-inoculant production and quality".

I consider it a topic of great relevance and interest both for research and for the formulation of products in the context in which organic farming is constantly expanding.

This paper is well designed and written, that's why I have only two recommendations:

1) Although each chapter including the one of conclusions also contains some recommendations for the development of this field as it is presented in the abstract (L30-31) I consider that a separate chapter regarding the development/perspectives of this field would be useful and more appealing to readers.

2) I consider that in the introduction should be listed the strategies that the title of the paper announces (Fermentation strategies to improve soil bio-inoculant production and quality) so that the reader can correlate the chapters of the paper with its purpose.

Author Response

Dear Reviewer, 

Thank you for your critical notes and suggestions: A separate chapter (modified Conclusions) was included at the end of the manuscript regarding the development/perspectives in the field; The fermentation strategies that the title announces were included in the Introduction.

We feel that after these corrections the MS is better.